# An Infrared Thermography Approach to Evaluate the Strength of a Rock Cliff

**Marco Loche** [1] , **Gianvito Scaringi** [1,*] , **Jan Blahůt** [2] , **Maria Teresa Melis** [3] , **Antonio Funedda** [3] , **Stefania Da Pelo** [3] , **Ivan Erbì** [3] , **Giacomo Deiana** [3] , **Mattia Alessio Meloni** [3] and **Fabrizio Cocco** [3]

1. Institute of Hydrogeology, Engineering Geology and Applied Geophysics, Charles University, Albertov 6, 128 43 Prague, Czech Republic; marco.loche@natur.cuni.cz
2. Institute of Rock Structure & Mechanics, Czech Academy of Sciences, V Holešovičkách 41, 182 09 Prague, Czech Republic; blahut@irsm.cas.cz
3. Department of Chemical and Geological Sciences, University of Cagliari, Cittadella Universitaria-S.S. 554 Bivio per Sestu I, 09042 Monserrato, Cagliari, Italy; titimelis@unica.it (M.T.M.); afunedda@unica.it (A.F.); sdapelo@unica.it (S.D.P.); ivanerbi@hotmail.it (I.E.); giacomo.deiana@unica.it (G.D.); melonimattiaalessio@tiscali.it (M.A.M.); fabrcocco@unica.it (F.C.)
* Correspondence: gianvito.scaringi@natur.cuni.cz

**Abstract:** The mechanical strength is a fundamental characteristic of rock masses that can be empirically related to a number of properties and to the likelihood of instability phenomena. Direct field acquisition of mechanical information on tall cliffs, however, is challenging, particularly in coastal and alpine environments. Here, we propose a method to evaluate the compressive strength of rock blocks by monitoring their thermal behaviour over a 24-h period by infrared thermography. Using a drone-mounted thermal camera and a Schmidt (rebound) hammer, we surveyed granitoid and aphanitic blocks in a coastal cliff in south-east Sardinia, Italy. We observed a strong correlation between a simple cooling index, evaluated in the hours succeeding the temperature peak, and strength values estimated from rebound hammer test results. We also noticed different heating-cooling patterns in relation to the nature and structure of the rock blocks and to the size of the fractures. Although further validation is warranted in different morpho-lithological settings, we believe the proposed method may prove a valid tool for the characterisation of non-directly accessible rock faces, and may serve as a basis for the formulation, calibration, and validation of thermo-hydro-mechanical constitutive models.

**Keywords:** compressive strength; infrared thermography; rebound hammer; cooling rate index

## 1. Introduction

The mechanical characterisation of rock masses has been the object of extensive research for decades [1,2]. Despite significant advances brought by improved instruments and methodologies, standard classifications—relying on the evaluation of rock strength and fracture networks [3]—remain widely utilised in direct field surveys [4–7]. The rock strength, defined as the resistance to permanent deformation by flow or fracture, is often estimated by the Schmidt (rebound) hammer test [8–14]. Direct surveying and ground-based monitoring, however, can be unfeasible if the rock outcrops are located in inaccessible areas, such as steep mountain ridges or coastal cliffs [15–18]. Efforts are therefore being made on formulating alternative methods relying on remote sensing techniques for the definition of input data for empirical and physically-based models [19–24].

Coastal cliffs are dynamic environments featuring frequent and rapid mass movements that can pose significant hazards to people, infrastructures, and ecosystems [25]. The stability of coastal cliffs is particularly sensitive to changes in hydro-meteorological forcing [26]. More frequent extreme weather conditions resulting from global warming may enhance physical weathering, instability phenomena, and cliff retreat in some regions [27]. Higher temperatures may also accelerate chemical weathering [27–29].

The use of unmanned aerial vehicles (UAVs) to perform detailed observations of non-directly accessible areas has become increasingly common, both to characterise rock mass geometries and to detect surface changes and movements [30–34]. Sensors operating in various ranges of the spectrum (Table S1) can be installed on board of UAVs, typically exploring the visible and infrared wavelengths in a similar fashion as the sensors installed on satellites, but without the limitations of fixed passing times and atmospheric shadowing. Depending on the desired precision and the extent of the study area, observations can also be performed from static locations on the ground.

Infrared thermography (IRT) is a remote sensing technique by which the surface temperature of a body can be evaluated from its thermal radiation [35–39]. Rocks behave as grey bodies, and the energy they emit follows Stefan-Boltzmann's law: $J = \varepsilon \theta T^4$, where $J$ is the total energy emitted by a body, $T$ is its surface temperature, $\varepsilon$ is the emissivity, and $\theta$ is Stefan-Boltzmann's constant [40]. Assessing the energy balance during a heating phase can be complex owing to the direct (and at times inconstant) exposure of the rock to solar radiation. Conversely, the evaluation $T$ during a subsequent cooling phase, that is when the body re-equilibrates with the ambient temperature, is simpler and more viable [41].

The potential of IRT in geosciences has been first demonstrated in the seminal work of Hudson [42]. Since then, several applications have been proposed thanks to the technological development of thermal sensors and acquisition systems [16]. Volcano monitoring [43–46], underground mining [36], cave exploration [47], and geothermal analyses [48] can benefit from IRT technologies. Landslide and rockfall mapping by IRT also have been attempted [37,39,49–54] (Table 1).

Thermal anomalies in rock masses can be related to the presence of loosened material and open fractures [55]. IRT monitoring during cooling can inform on the degree of fracturing in the field [41] and porosity in the laboratory [56,57]. These characteristics also affect the mechanical strength [3]. Therefore, the use of IRT to evaluate rock mechanical properties can be hypothesised. Indeed, here we present preliminary results of an IRT application to the prediction of the compressive strength of rock blocks in a portion of a 30-m high landslide-prone coastal cliff (Cala Delfino, south-east Sardinia, Italy; Figure 1). We demonstrate that the cooling trend of the rock blocks can be related with conventionally-evaluated rock strength values, and we define a cooling rate index (CRI) to obtain quantitative predictions through regression analysis.

**Table 1.** Features investigated by IRT in recent studies.

| Ref. | Platform (Distance Object-Sensor) | Features |
|------|-----------------------------------|----------|
| [58] | Terrestrial | Eroded caves in a shotcreted slope |
| [37] | Terrestrial (120–150 m) | Shallow inhomogeneities, weathered rock cliff areas |
| [23] | Terrestrial | Main joints, recently collapsed areas/detachments in a coastal cliff |
| [47] | Terrestrial, UAV | Open cracks, tension and loosened zones, pseudo–karst caverns |
| [41, 55] | Terrestrial (3 m) | Geostructural features, fracturing degree, daytime temperature exchange of a rock slope |
| [39] | Terrestrial | Thermal contrast between vegetated, weathered and bare rock areas of an unstable slope |
| [59] | Terrestrial | Discontinuity system of a rock wedge |
| [51] | Terrestrial, airborne | Wedge fractures, erosional channels, scarps, earthflow ponds, seepage sectors, debris cones |
| [60] | Terrestrial | Surficial temperature, thermal response of jointed blocks, seasonality |
| [61] | Terrestrial (20 m) | Spatio-temporal surficial temperature pattern of a rock mass arch |
| [53] | Terrestrial (600 m) | Weathering rock areas, moisture content related to the ephemeral drainage network |

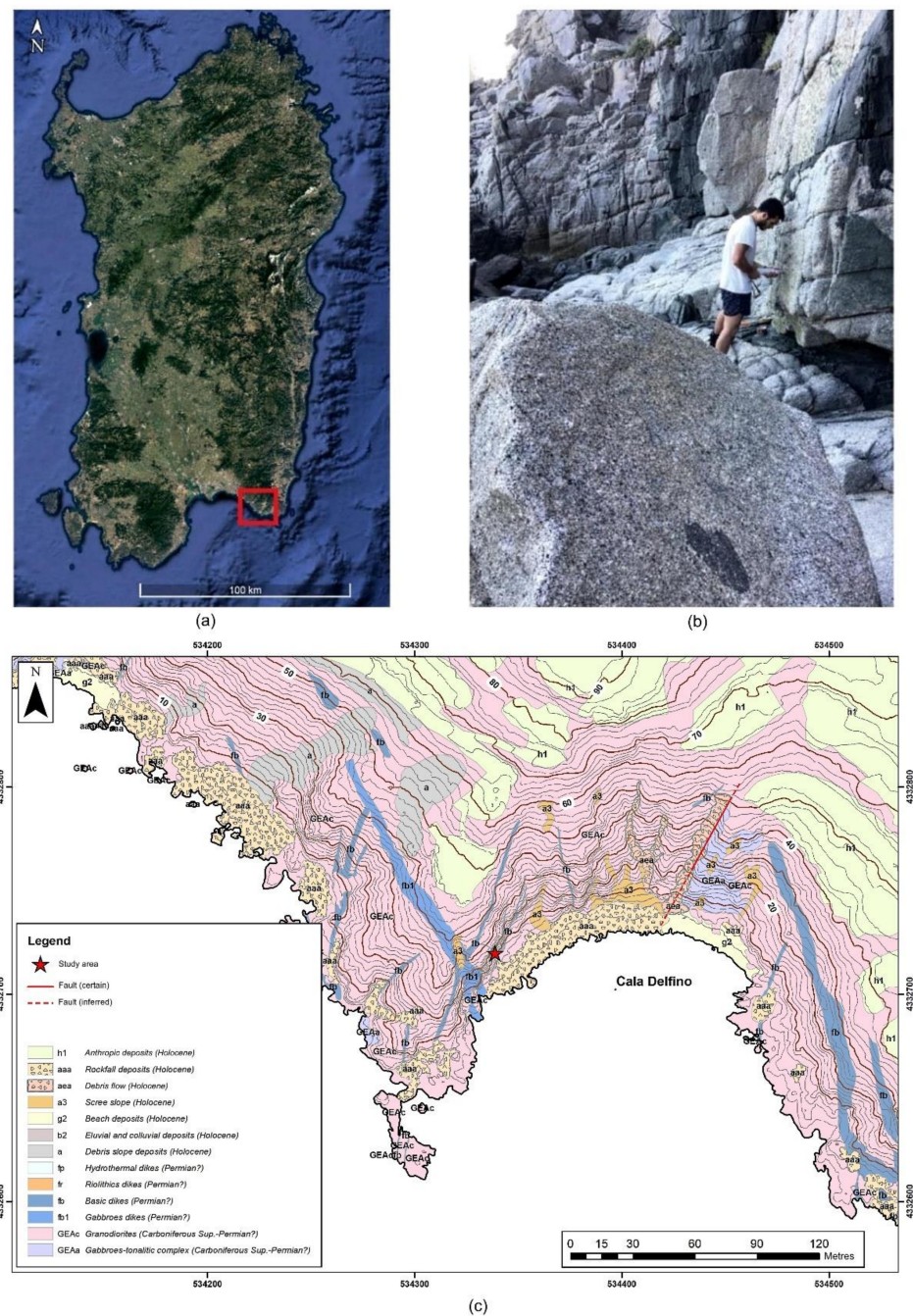

**Figure 1.** (**a**) Location of the study area; (**b**) rebound hammer test; (**c**) geological map. The plutonic rocks belong to the Geremeas intrusive Unit (GEA) [62]. Coarse-grained, equigranular to moderately inequigranular Granodiorites (GEAc) are present, with medium-fine grained, greenish-grey enclaves of gabbro-tonalitic masses (GEAa). The dikes crosscutting the unit are generally vertical, 1–10 m thick, NNW- and, subordinately, NE-trending. They are basaltic to andesitic dikes (fb), grey-green to blackish, with aphanitic to microcrystalline or porphyritic structure for phenocrysts of feldspar and/or amphibole, and gabbroid dikes (fb1), dark grey, with microgranular structure for plagioclase, pyroxene and quartz. The stabilised slope deposits (**a**) consist of debris accumulations of angular clasts, locally with matrix, at times partially compacted and stabilised. Eluvial-colluvial deposits (b2) are made of debris in a fine matrix, sometimes intercalated with more or less evolved soils, organic-enriched. Beach deposits (g2) are made of current sands, gravels and pebbles with local remains of Posidonia oceanica. The active slope deposits (a3) are still evolving, chaotic accumulations of incoherent clasts in a finer matrix, mainly of weathered granitoids. Rockfall deposits (aaa) consist of dm- to m-sized angular blocks. Debris flow deposits (aea) are unsorted, chaotic angular clasts, blocks, plant remains, with some anthropic material in a fine sandy-silty matrix. Anthropic deposits (h1) from manufactures and filling materials are also present.

## 2. Site Characterisation

The study site is located in the Variscan crystalline basement of south-east Sardinia, Italy (Figure 1), and is part of the Sàrrabus igneous massif (400 km$^2$), a composite intrusive complex related to the late phase of the Variscan orogeny, emplaced between 305 Ma and 285 Ma in the frontal part of the orogenic wedge [63]. Here, the crystalline basement is characterised by two lithofacies of plutonic rocks intruded by a Permian dike complex, both covered by thin Quaternary, continental to littoral sediments. These consist of eluvium-colluvial layers, slope deposits, debris flow and rockfall deposits, contemporary beach deposits, and small anthropic deposits. Rockfall deposits are widespread at the base of the cliffs and along the small bays. They consist of angular blocks of variable size (dm$^3$ to m$^3$); the largest blocks are mainly from the granodioritic and tonalitic lithofacies.

The basements rocks exhibit a widespread fracture network, where two different systems—related to the late Variscan evolution—can be distinguished and often bear ore deposits in adjacent areas [64]. At the map scale, the NNW-trending fracturing dominates, and affects the whole Sàrrabus region for more than 50 km to the north [65]. The main Permian dikes follow this trend. A NE-trending system is also present, with minor dikes. At the outcrop scale, some other systems can be seen, with attitudes from vertical to sub-horizontal. The latter are joints related to the after-emplacement cooling of the intrusive bodies and their post-orogenic exhumation. In adjacent areas, evidence exist that these fractures have been reactivated during the Tertiary and Plio-Pleistocene evolution [66].

The juxtaposition of outcropping lithologies with different compositions makes the use of IRT favourable for characterising their properties. Within this context, we performed a 24-h long thermal monitoring of a cliff that borders the tourist beach of Cala Delfino. Along the cliff, the size of the rock blocks ranges from less than one metre to several metres (Figure 1b). We selected three blocks with dimensions of 2 m × 0.5 m (block A), 1 m × 1 m (block B), and 0.5 m × 0.5 m (block C) (Figure 2a). Macroscopically, the blocks present either uniform or no fracturing. Blocks A and B consist of granodiorites, while the darker block C consists of basic glass-free hypocrystalline aphanitic rocks. In thin sections, quartz and chloritised biotite in the groundmass were observed. Several amphiboles and large plagioclase crystals were also found. Micro-fractures systems were observed, marked by chlorite or iron oxides. According to the volumetric joint count (Jv; [67]), defined as the number of joints intersecting a volume of 1 m$^3$, the blocks were attributed to three different classes (Table S2): very low Jv—massive block with no visible fractures (block A); low Jv—massive block with small fractures (block C); moderate Jv—continuous block with regular fracturing degree (block B).

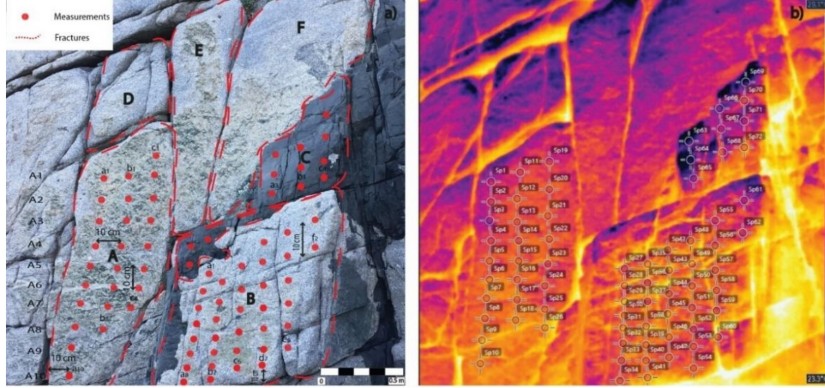

**Figure 2.** Acquisition scheme (**a**) in the field, where 10 rebound tests were performed at each point, and (**b**) in FLIR Tools software, where each point corresponds to a spot measurement within the thermogram. The image in (**b**) was acquired at 22:00.

## 3. Materials and Methods

The rock strength ($\sigma_c$) was evaluated through rebound hammer tests, as a strong correlation [8,14] can be properly estimated in the laboratory following conventional procedures [9,10,68]. Figure 2a displays the points where the tests were performed following ASTM recommendations. Ten readings were taken at each point (red dots: e.g., a1, b1, c1). Considering that $\sigma_c$ is strongly influenced by the density, distribution and connectivity of the microstructures, in order to reflect the potential spread of the values resulting from rock heterogeneity, block-level mean, median, mode and standard deviation values were also computed. The temperature of each point in each thermogram was also evaluated, so that the surface temperature of each block through time could be assessed and its trend related with the rock strength.

Thermal data were acquired during the summer season (from 12:00 p.m. on 23 June 2020 to 12:00 p.m. on 24 June 2020) through a drone-mounted FLIR XT2 camera (Table S3). The drone, in our specific application, was installed on a fixed platform to ensure stability even under a breeze, and continuous monitoring for the desired period of time (Figure 3a). The shooting point was located about 5.4 m from the rock face to record images with sufficient detail at an approximately perpendicular angle. After processing, each pixel of the recorded image (e.g., Figure 2b) represented a value of surface temperature.

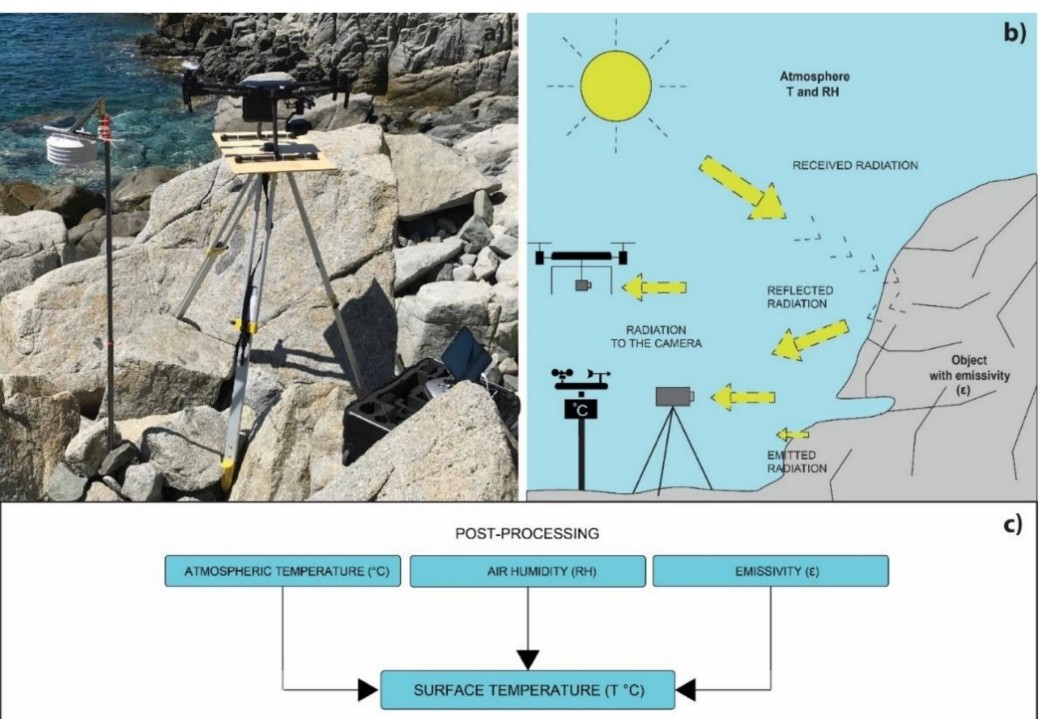

**Figure 3.** (**a**) Drone and weather station; (**b**) schematic of the acquisition system (a ground-based solution was adopted in this work) and heat transfers between the cliff and the atmosphere [16]; (**c**) post-processing scheme.

The blocks were directly exposed to solar radiation during the morning (7–12 h) given their south-east exposure. In the afternoon and night (13–7 h), they gradually released the heat into the environment (Figure 3b). Thermograms were acquired at 60-min intervals for 24 h. Air humidity and temperature were recorded through a weather station (Table S4) to calibrate the thermograms during image post-processing. (Figure 3c). Emissivity was assumed according to Öhman [69].

The digital elaboration of the IRT images was carried out using FLIR Tools (FLIR Systems, Inc. Wilsonville, OR, USA), a software package designed to edit radiometric images that allows the evaluation of temperatures at specific points, along lines, or over

areas. Point and line measurements were used to define individual blocks and fractures, respectively. The temperature of a block surface was estimated by averaging the point measurements, which were taken over a regular grid corresponding to that used for the rebound tests (Figure 2). In this way, the difference of temperature within the same thermogram could be highlighted, and specific temperature ranges could be isolated to define the best representative output. The average temperature of the block face was calculated by a point function, allowing a weighted measure of the face for each thermogram for 24 h. The resulting values were elaborated to determine the cooling and heating phases (Figure 4).

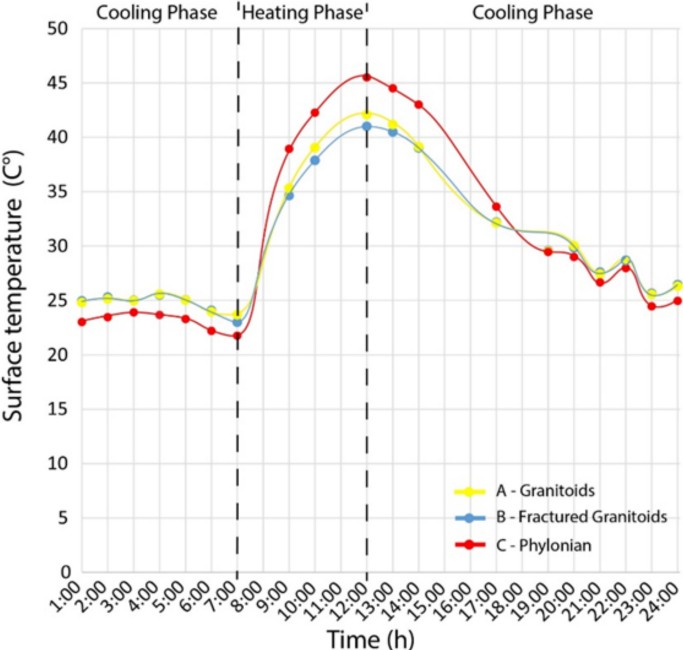

**Figure 4.** Cooling and heating phases computed from thermograms over 24 h.

Different rocks may express different cooling properties, which we characterised in terms of a cooling rate index ([41,57]) that quantifies the variation of temperature ($\Delta T$, °C) per unit time ($\Delta t$, h) as: CRI = $\Delta T / \Delta t$.

## 4. Results and Discussion

We calculated the mean, median, mode, standard deviation (SD) and range of the rebound values for each block (Table 2).

**Table 2.** Rebound number statistics of the assessed blocks.

| Block | ASTM | Mean | Median | Mode | SD | Range |
|-------|------|------|--------|------|-----|-------|
| A | 53 | 53 | 54 | 58 | 9 | 18–69 |
| B | 49 | 49 | 50 | 52 | 11 | 18–68 |
| C | 60 | 59 | 60 | 62 | 5 | 35–68 |

With the ASTM method, three extreme values are discarded for each point, and the remaining values are averaged to give the point rebound number, then averaged again to give the block rebound number. The mean is computed over all readings on each block (10 readings per point, 10 points per block). Median, mode, standard deviation and range also are computed over all readings. A slight asymmetry in the dataset can be recognised from the median and modal values being larger than the mean. However, differences among the statistics are minor, and the estimated compressive strength values do not differ much either. Using the ASTM method and the empirical relationship provided in the hammer

specifications ($\sigma_c = 0.0232 \times R^{2.2637}$, where R is the rebound number), the following strength values are obtained: block A, 186 MPa; block B, 156 MPa; block C, 246 MPa. Using the mean of all measurements, the strength of block C would be 237 MPa, while those of blocks A and B would remain unchanged. The use of median or modal values would lead to 12–23% higher strength estimates. Uncertainty ($\pm 1$ SD) on the strength is 36% on average. This may seem a large value, yet we will demonstrate that it does not hinder a strong correlation with the CRI.

Physically, the different strength of blocks A and B, notwithstanding the similar lithology, can be attributed to the different degree of fracturing. Following Deere & Miller's classification [70] (Table S5) block C can be classified as a very hard rock while blocks A and B as hard rocks.

The thermograms captured the heating-cooling cycle of the cliff during a 24-h period. By comparing images taken during the day and during the night (Figure 5), it is apparent that different zones are heated and cool differently. In particular, clean faces and corners (according to the diffusion equation) are the first surfaces to lose the heat adsorbed during the day. The thermal behaviour of fractures is likely influenced by air circulation within them, and therefore may be informative, to some extent, of the internal thermal state of the rock mass [41]. In fact, fractures appear relatively cooler during daytime and relatively warmer during the night. The morphology and structure of the blocks also can influence their heating-cooling behaviour. Block C, constituted by basic rock (more similar to a black body) with weak degree of fracturing, tends to lose the heat more homogeneously than the other blocks.

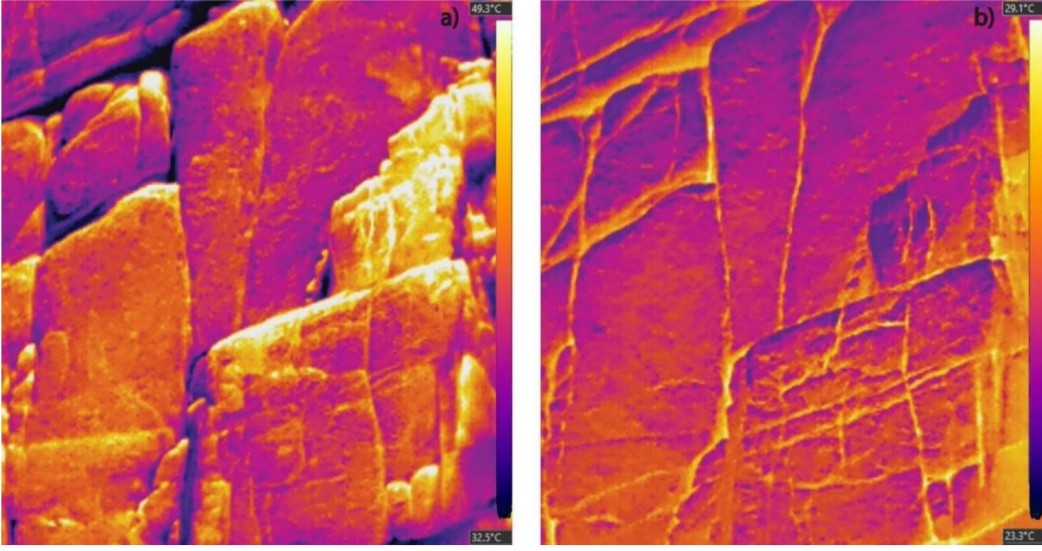

**Figure 5.** Thermograms recorded at (**a**) midday and (**b**) midnight where small fractures show less heat compared with larger ones.

To obtain a quantitative insight, we elaborated the images numerically, and estimated temperature values at the locations where the rebound hammer test was performed. The blocks located in unreachable locations and the discontinuities were also analysed. The average temperature of each block or discontinuity in each thermogram was evaluated and plotted against time (Figure 6). With reference to the day-night cycle, the temperature increased rapidly on the block faces during the morning hours under direct exposure to solar radiation and reached a peak at 12:00 p.m.; then, the rock gradually cooled throughout the afternoon and evening, and the temperature remained substantially stable from midnight to the following morning. The heating-cooling pattern was similar for all blocks. However, the amplitude of the thermal oscillation was different: it was larger for block C (aphanitic) than for blocks A and B (granitoids).

The results support the hypothesis that the cooling behaviour of rock surfaces can be related to their inner macroscopic structure. The cooling phase is simpler to model than the heating phase as it occurs with smaller environmental disturbance. Conversely, in the morning hours, the heat balance depends on both the solar irradiance (direct and reflected by surrounding bodies) and the heat transfer from the rock to the atmosphere (generally cooler) supported by air convection. The rate of change in the temperature of a body is proportional to the difference with the ambient temperature. A faster cooling rate can therefore be indicative of a higher body temperature, provided that heat capacity and conductivity remain the same. Different structures (e.g., massive blocks or fractured rocks) can exhibit different cooling behaviours in relation to the different surface available for heat exchange, the role played by air convection, but also the insulating effect of the latter. In fact, air convection seems to prevail in our study case: the massive block C cools faster than block B, which has strong fracturing and different lithology (Figure 6). The thermal behaviour of the fractures differs from that of the blocks: lower temperatures during the day and higher temperatures during the night are recorded compared with those of the blocks. This signals an insulating effect of the air, and a role played by convection in keeping the inner part of the rock relatively cooler during daytime and warmer during the night. Furthermore, we observe that small fractures retain less heat compared with the larger ones (Figure 5). This may allow for an IRT-based hierarchisation of fractures.

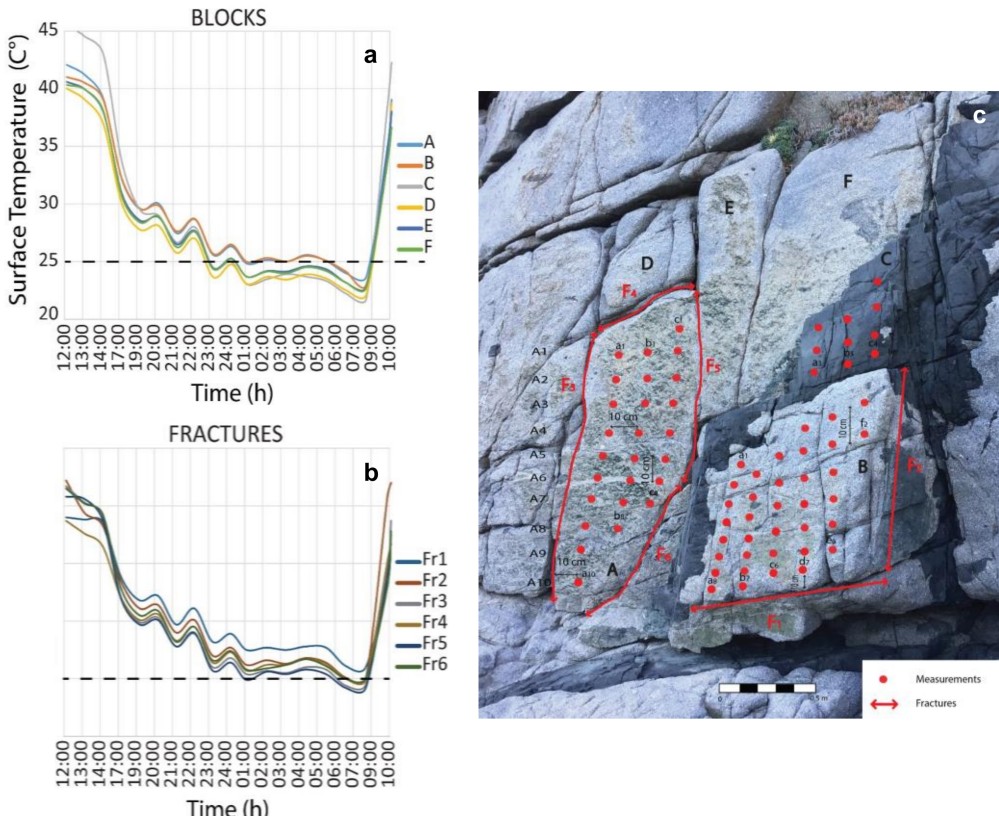

**Figure 6.** Cooling and heating patterns of (**a**) blocks and (**b**) fractures computed by thermograms. Blocks and fractures are shown in (**c**): Fr1 and Fr2 are open fractures with a length of 0.6 m; Fr3 is a semi-open fracture with a length of 1.2 m; Fr4 is an open fracture with a length of 0.4 m; Fr5 is a semi-open fracture with a length of 0.7 m; and Fr6 is a semi-open fracture with a length of 0.6 m.

We performed a regression analysis between the estimated rock strength and the CRI of blocks A, B, and C. A positive correlation between CRI and Jv has been highlighted in the literature in smaller domains, without focusing on block-scale mechanical features or performing systematic temporal data acquisition [41]. Here, we explored different time intervals to define the CRI. Qualitatively, the different cooling behaviours of the blocks

are particularly evident in the first hours of cooling (i.e., from 12:00 to 17:00), when the thermal gradients are the strongest. In fact (Figure 7) we found the highest coefficient of determination ($R^2 > 0.99$) between the rock strength and the CRI evaluated over the first five hours of cooling ($CRI_{5h}$, 12:00–17:00). Smaller correlations are evaluated for shorter time intervals, i.e., with $CRI_{1h}$ (12:00–13:00, $R^2 = 0.75$) and $CRI_{2h}$ (12:00–14:00, $R^2 = 0.13$). A strong correlation is also seen with the Total CRI ($R^2 > 0.97$), which expresses the total thermal excursion from midday to the following morning (12:00–07:00). The identification of $CRI_{5h}$ as the most informative index has a practical implication as it suggests that a 24-h thermal monitoring would be superfluous and could be replaced by a shorter monitoring period to cover only the first hours of cooling, which are associated with the dissipation of the strongest thermal gradients. However, this finding needs to be confirmed on different lithologies, structures, and spatial scales.

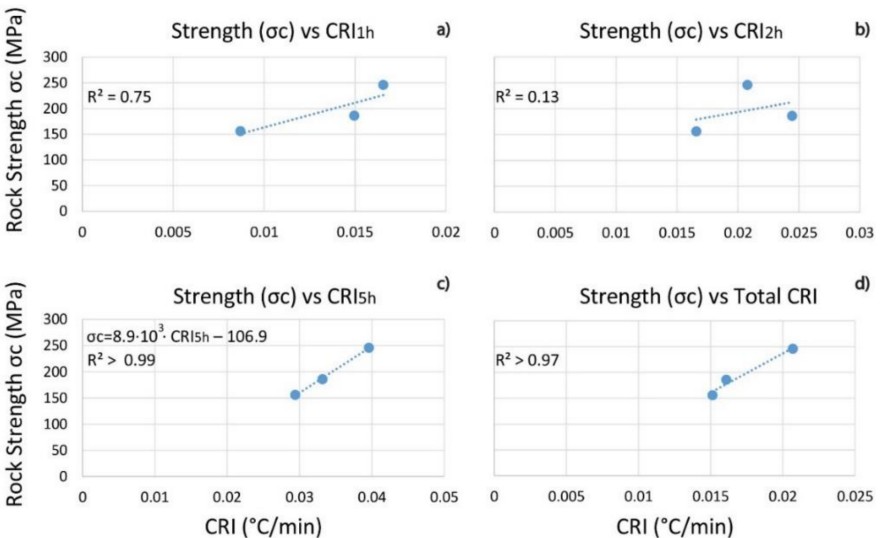

**Figure 7.** Linear regression analysis between rock strength ($\sigma_c$) and CRI according to different time intervals since the beginning of cooling: (**a**) 1 h; (**b**) 2 h; (**c**) 5 h; (**d**) entire cooling phase. The markers represent the mean values for blocks A, B and C.

We evaluated the robustness of our correlations by performing a Monte-Carlo simulation (10,000 independent repetitions). To reproduce the actual measuring process, we drew 10-plets of rebound values as well as values of temperatures for each measuring point in each of the three blocks at each measuring time (300 rebound values and 30 temperature values at each measuring time). We imposed normal distributions for all variables, derived from the empirical values (Table 3).

We then proceeded to calculate the mean rebound numbers and temperatures, and thus the strength and CRIs for each block, and analysed the distribution of $R^2$ values resulting from the linear regressions for each repetition. We obtained median $R^2$ values of 0.67, 0.40, 0.93, and 0.92 between $\sigma_c$ and $CRI_{1h}$, $CRI_{2h}$, $CRI_{5h}$, and Total CRI, respectively. We calculated selected percentiles of the distributions of $R^2$ and slope values (m) of the regression (Table 3), based on which we can infer a positive correlation (m > 0) between $\sigma_c$ and $CRI_{5h}$ (*p*-value < 0.01) and between $\sigma_c$ and Total CRI (*p*-value < 0.001). For the $\sigma_c$—$CRI_{5h}$ and $\sigma_c$—Total CRI regressions we can also infer (*p*-value < 0.05) that $R^2 > 0.44$ and $R^2 > 0.50$, respectively. In other words, we can affirm that the thermal behaviour, through 5-h (or total) cooling rate index, can account for at least 44% (or 50%) of the variance observed in the compressive strength. On the other hand, we cannot infer with sufficient confidence the existence of a positive correlation between $\sigma_c$ and $CRI_{1h}$ (*p*-value > 0.05), or between $\sigma_c$ and $CRI_{2h}$ (*p*-value > 0.05). Indeed, at *p*-value = 0.05, the data appear uncorrelated ($R^2 < 0.02$) in both cases.

**Table 3.** Input and output of a Monte-Carlo simulation (10,000 independent repetitions) to assess correlations between rebound number-derived compressive strength and cooling rate indices with different time spans. $R^2$ and m are the coefficient of determination and the slope of the linear regression between $\sigma_c$ and CRI (i.e., $\sigma_c = m \cdot CRI + q$); perc. = percentile.

| Input | | | | | | | | | | | | |
|---|---|---|---|---|---|---|---|---|---|---|---|---|
| **Block** | **Rebound** | | **T at t = 12:00** | | **T at t = 13:00** | | **T at t = 14:00** | | **T at t = 17:00** | | **T at t = 7:00** | |
| | **Mean** | **SD** | **Mean** | **SD** | **Mean** | **SD** | **Mean** | **SD** | **Mean** | **SD** | **Mean** | **SD** |
| A | 53 | 9 | 42.09 | 0.90 | 41.19 | 0.75 | 39.15 | 0.55 | 32.13 | 0.45 | 23.74 | 0.43 |
| B | 49 | 11 | 41.02 | 0.74 | 40.50 | 0.70 | 39.03 | 0.45 | 32.20 | 0.54 | 23.78 | 0.43 |
| C | 60 | 5 | 45.51 | 1.53 | 44.52 | 1.89 | 43.02 | 2.20 | 33.64 | 0.65 | 21.92 | 0.99 |

| Output | | | | | | | | |
|---|---|---|---|---|---|---|---|---|
| **Statistic** | **$R^2$ of $\sigma_c$ vs.:** | | | | **m of $\sigma_c$ vs.:** | | | |
| | **CRI$_{1h}$** | **CRI$_{2h}$** | **CRI$_{5h}$** | **Total CRI** | **CRI$_{1h}$** | **CRI$_{2h}$** | **CRI$_{5h}$** | **Total CRI** |
| 0.1 perc. | 0.00 | 0.00 | 0.01 | 0.02 | <0 | <0 | <0 | 113 |
| 1st perc. | 0.00 | 0.00 | 0.13 | 0.24 | <0 | <0 | 1908 | 4259 |
| 5th perc. | 0.01 | 0.00 | 0.44 | 0.50 | <0 | <0 | 3850 | 7576 |
| mean | 0.59 | 0.45 | 0.86 | 0.86 | 1972 | 2557 | 8969 | 15227 |
| median | 0.67 | 0.40 | 0.93 | 0.92 | 2663 | 2768 | 8628 | 15167 |

The observations involving CRI$_{1h}$ and CRI$_{2h}$ are certainly affected by the comparatively small decrease of temperature (with respect to its spatial variability) evaluated in the first two hours of cooling. In other words, the excessive "signal-to-noise" ratio hinders conclusions in short time spans. Conversely, as already mentioned, there is little difference between the inferences for CRI$_{5h}$ and the Total CRI. The former can, in a time span of five hours (29% of the total cooling time), account for about half (50–54%) of the total cooling. Therefore, the use of CRI$_{5h}$ should be preferred owing that it can be calculated after a much shorter monitoring period.

## 5. Conclusions

We have identified a strong correlation between the cooling rate index of rock blocks in a coastal cliff prone to rockfalls—evaluated through infrared thermography—and their compressive strength—evaluated by the rebound hammer test. Such correlation could be useful for characterising cliff portions that are unreachable for direct geomechanical surveying. The identification of the first five hours of cooling (after the temperature peak at midday) as the most informative time interval for regression analysis suggests that thermal sensing can be more effectively performed in the afternoon hours rather than over a complete 24-h cycle. The quantitative analysis of the thermograms can also highlight different heating-cooling patterns related to different lithologies, structures, and fracture sizes/apertures. Therefore, it can constitute a valid aid in rock and fracture network classifications. Infrared thermography can be flexibly applied at various spatial scales: few repeated flights of an UAV equipped with a thermal camera, performed during a cooling phase, can be sufficient to characterise the strength of unreachable blocks after establishing an empirical correlation based on field strength measurements. In principle, thermal information obtained by spaceborne sensors could also be related to that obtained by UAVs and ground-based sensors for studies with larger spatial coverage. However, in such cases, the limitations related to the achievable resolution, the opacity of the atmosphere to certain wavelengths, and the fixed passing times of the satellites should be addressed. With respect to the investigation of limited portions of a cliff, such as in the case study presented herein, the proposed methodology appears solid. However, further validation is needed in different settings (climates, seasons, meteorological conditions, lithologies, geometries), so as to explore multi-variate correlations applicable over larger areas, and

aid in the formulation, calibration, and validation of thermo-hydro-mechanical models of rock faces.

**Supplementary Materials:** The following are available online at https://www.mdpi.com/article/10.3390/rs13071265/s1. Table S1: Primary spectral regions used in remote sensing. Table S2: Degree of fracturing based on the volumetric joint count). Table S3: Specifications of the FLIR XT2 camera. Table S4: Data from the weather station, external and internal data of the station, and laser thermometer data. Table S5: Strength classification of rocks.

**Author Contributions:** Conceptualisation and methodology, M.L., G.S., J.B., M.T.M., A.F., S.D.P., I.E., G.D. and M.A.M.; formal analysis, M.L., G.S. and J.B.; investigation, M.L. and I.E.; data curation, and original draft preparation, M.L.; review and editing, M.L., G.S., J.B., M.T.M., A.F., S.D.P., G.D., M.A.M. and F.C.; visualisation, M.L.; supervision, G.S. and M.T.M.; project administration, A.F. and G.S., funding acquisition, G.S., M.T.M., A.F. and S.D.P. All authors have read and agreed to the published version of the manuscript.

**Funding:** The experimental work was funded by Project MAREGOT under the Program: 2014–2020 INTERREG V-A Italy–France (Maritime). The first author appreciates the financial support given by the Charles University Grant Agency (GAUK) with project number 337121. Data analysis, manuscript preparation and publication were funded by the Grant Agency of the Czech Republic (GAČR Grant No. 20-28853Y) and the Fund for international mobility of researchers at Charles University (MSCA-IF IV; Project No. CZ.02.2.69/0.0/0.0/20_079/0017987).

**Institutional Review Board Statement:** Not applicable.

**Informed Consent Statement:** Not applicable.

**Data Availability Statement:** Elaborated data are presented in the manuscript. Raw experimental data can be provided by the authors upon reasonable request.

**Conflicts of Interest:** The authors declare no conflict of interest. The funders had no role in the design of the study; in the collection, analyses, or interpretation of data; in the writing of the manuscript, or in the decision to publish the results.

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
