# Peer review of "An Infrared Thermography Approach to Evaluate the Strength of a Rock Cliff"

_remotesensing, doi:10.3390/rs13071265_

Round 1
Reviewer 1 Report
The objective of this submission is to report on the novel method dealing with evaluation of the compressive strength of rock blocks based on the thermal behaviour monitoring by infrared thermography. It is found that the lithology and structure of the rock blocks and to the size influence a lot on the heating-cooling patterns. It would be nice to see not only experimental studies (which is very interesting and important), but also mathematical model confirmed with the experiments.
Author Response
We appreciate the reviewer taking the time to provide constructive feedback on our manuscript. Our study is indeed an experimental one, and we wished to present these preliminary results to stimulate further research in the developing field of thermal remote sensing. Some of us have participated in the preparation and publication of a review paper on the topic (https://doi.org/10.3390/rs12121971), where the theoretical framework was laid out, supported by laboratory experiments and first field experiences.
Based on our experimental results, we demonstrated how to construct a simple empirical relationship that provides a practical mean to estimate strength values from the cooling rate index which, in turn, is built upon remotely-sensed values of emissivity translated into temperatures through the Stefan-Boltzmann equation.
Clearly, such an empirical relation can only be used in the specific context (e.g., adjacent blocks, similarly oriented, with the same lithology and fracturing degree). However, the empirical approach has a general validity and the experiment could be (and should be) repeated in other morphological, lithological, and climatic settings to identify regularities that would allow the formulation of multi-variable empirical equations with more general applicability.
Given the number of variables at play in the heating-cooling pattern of rock blocks, we believe that it would make sense to compare experimental results with physically-based (constitutive and numerical) modelling results only once a large amount of data from diverse settings and the associated correlations are available. This is certainly very stimulating and is indeed part of our research roadmap. We edited the manuscript accordingly to better reflect the above points.
Reviewer 2 Report
In their work, authors carried out experiments to establish a relationship between IRT (Infrared Thermography) images and strength of rock blocks so that a drone mounted IR camera can be used to identify the rock strength remotely. To evaluate the rocks’ strength, they evaluate the cooling behaviors of the rock blocks (in this case the identification of CRIs) by using IRT images during the natural cooling period. They then correlate these values with the strength of the rocks calculated from the rebound hammer tests data.
There are two main issues I would like to raise about the methodology; first one is the experimental set up in situ: as I mentioned on the manuscript, selected rock surfaces are not in the same plane (block C has an angle to the plane) and this creates the shadow/cooler surfaces that we observe in the IRT images of Block C. Since that kind of out of plane form may result in the different cooling behavior. To evaluate the efficiency of the methods it is better either to limit the parameters for the interpretation of the results that is to say to select a block in the same plane or to give an explanation why it would not create a problem for the interpretation of the IRT images and cooling behavior of this block.
Furthermore, is there any lab work done on the thermal behavior of the rocks, in other words have you followed the cooling behavior of these rocks with IRT camera in controlled lab conditions? I think it is worth trying to create a controlled environment in the lab and follow the cooling/heating behavior of the rocks first and then interpreting the results in situ would be more efficient and reliable. Placing a control sample during data acquisition might have been a way to get around this problem.
In Figure 4. the consequent heating and cooling phases are seen. Can heating rate also be used for the estimation of the strength just as cooling rate? As it can be seen from the graph a heating period between 9:00 ad 12:00 is clearly different for all the rocks. In other words, why cooling is more advantageous over heating? An explanation for the preference is highly appreciated since this would affect the time spent for the data collection.
In the introduction section in line 30-31 the references should be given in order of appearance in the text. Also, the references should be organized accordingly.

Author Response
We greatly appreciate the reviewer for taking the time to go through our manuscript and provide constructive comments.
Concerning the first point - non planarity of the studied surfaces generating shadows and affecting the heating-cooling pattern, we believe the reviewer misinterpreted the image in Fig. 2a as block C is simply much darker (different lithology) and not in shadow. The planarity of the block surfaces can be observed in Fig. 1b. The different color in the thermogram in Fig. 2b does not relate to any shadows (the image was aquired at 22:00 - this has now been clarified in the caption) but to an actual difference in the cooling pattern.
As for the laboratory experiments, indeed, we did not perform them within the scope of the project funding our research, which was meant to provide a field verification of past laboratory and theoretical results. We followed previous works (e.g., Mineo, S., et al., 2005, doi:10.3301/ROL.2015.103), which showed, albeit on different materials, strong correlations in laboratory conditions. Of course, we agree with the reviewer that the replication of the observed pattern in controlled conditions on the very same materials is important and can provide better insight into the physical process of heating-cooling in relation to a number of rock properties, including their mechanical strength. In fact, we are finalizing in these days the submission of a more comprehensive grant proposal which would allow us to carry out such experiments in the laboratory and in the field on multiple lithologies and environmental conditions, and use the generated results to formulate a thermo-hydro-mechanical model with more general validity.
Furthermore, the reviewer asks whether the heating phase also could be used to extract relevant information for strength estimation. This is true in principle. However, the energy balance during heating is more complex as it also depends on the direct (and at times inconstant) exposure of the rock to solar radiation. As already judged in a previous publication (Pappalardo et al., 2016, doi:10.1016/j.ijrmms.2016.01.010), the interpretation of the cooling phase, in which the rock behaves as a grey body emitting radiation while re-equilibrating with the ambient temperature, is simpler and thus more viable.
Reply to minor comments:
- We reordered the references, justified the text, and correctly defined the "cooling rate index" (CRI) after its first occurrence.
Reviewer 3 Report
Dear Authors,
the manuscript titled "An infrared thermography approach to evaluate the strength of a rock cliff" presents an interesting approach to evaluate a mechanical parameter of rock through the measurment of the surface temperature of the rock.
The topic is of great interest and well introduced and discussed. Minor revision are required:
1)Fig. 6 It could be useful a picture of the rock face in which the analysed fractures are highlighted.
2)in the Results and discussion section (lines 265-269), the method for the evaluation of the robusteness (is it a bootstrap model?) has to be better clarifierd and detailed. As an example, I suppose that the 10-plets of rebound for 10 point are extracted, for each repetition, among the values of the normal distribution of rebound imposed for each block (with mean value and SD from table 3), but the sentence is not so clear. Moreover, why do you speak also about 10-plets of temperatures? For each of the 10 points, at each time, you extract 10 rebound values, but why 10 temperatures? Please clarify this paragraph.
Author Response
We appreciate the reviewer taking the time to go through our manuscript and provide constructive feedback.
The reviewer suggested that we modify Fig. 6 to better highlight the location of the fractures for which we show the heating/cooling trend. We agree with the reviewer and we have modified the figure accordingly.
Furthermore, the reviewer pointed out that the description of the Monte-Carlo simulation was somewhat unclear. The sentence actually contains a mistake and we apologise for this. The reviewer interpreted our sentence correctly. In order to simulate the actual process of sampling, we took 10 values of rebound at each point, 10 points per block for each of the 3 blocks. On the other hand, we took only one value of temperature at each point as only one image was acquired at each time and the spatial-temporal variability in the time span of the 10 rebound measurements at the same location is likely very limited. The correct sentence now reads: "We evaluated the robustness of our correlations by performing a Monte-Carlo simulation (10,000 independent repetitions). To reproduce the actual measuring process, we drew 10-plets of rebound values as well as values of temperatures for each measuring point in each of the three blocks at each measuring time (300 rebound values and 30 temperature values at each measuring time)."
Round 2
Reviewer 1 Report
This version can be accepted.
Author Response
We thank once again the reviewer for the constructive comments that helped improve our manuscript.
Best regards,
Gianvito Scaringi and coauthors
Reviewer 2 Report
My only small request would be adding authors explanation on heating phase given in their response to the reviewers in the text with one or two sentences.
As authors explained: "However, the energy balance during heating is more complex as it also depends on the direct (and at times inconstant) exposure of the rock to solar radiation. As already judged in a previous publication (Pappalardo et al., 2016, doi:10.1016/j.ijrmms.2016.01.010), the interpretation of the cooling phase, in which the rock behaves as a grey body emitting radiation while re-equilibrating with the ambient temperature, is simpler and thus more viable."
Figure captions are more clear and gives more explanation about the pictures.
Thank you very much...
Author Response
We thank the reviewer once again.
We have expanded a paragraph in the introduction (lines 53-60) which now reads: "Infrared thermography (IRT) is a remote sensing technique by which the surface temperature of a body can be evaluated from its thermal radiation [35–39]. Rocks behave as grey bodies, and the energy they emit follows Stefan-Boltzmann’s law: J=εθT4, where J is the total energy emitted by a body, T is its surface temperature, ε is the emissivity, and θ is Stefan-Boltzmann’s constant [40]. Assessing the energy balance during a heating phase can be complex owing to the direct (and at times inconstant) exposure of the rock to solar radiation. Conversely, the evaluation T during a subsequent cooling phase, that is when the body re-equilibrates with the ambient temperature, is simpler and more viable [41]."